



# Velocity of magnetic holes in the solar wind from cluster multipoint measurements

Henriette Trollvik[1], Tomas Karlsson[1], and Savvas Raptis[1]

[1]KTH Royal Institute of Technology, School of Electrical Engineering and Computer Science, Division of Space and Plasma Physics, Stockholm

**Correspondence:** H. Trollvik (trollvik@kth.se)

**Abstract.** We present the first statistical study on the velocity of magnetic holes (MHs) in the solar wind. Magnetic holes are localized depressions of the magnetic field, often divided into two classes; rotational and linear MHs. We have conducted a timing analysis of observations of MHs from the Cluster mission in the first quarter of 2005. In total, 69 events were used; out of these, there were 40 linear and 29 rotational MHs, where the limit of magnetic field rotation was set to $50°$. The resulting median velocity was $7.4 \pm 45$ km/s and $25 \pm 42$ km/s for linear and rotational MHs, respectively. For both classes, around 70% of the events had a velocity in the solar wind frame that was lower than the Alfvén velocity. Therefore, we conclude that within the observational uncertainties, both linear and rotational MHs are convected with the solar wind.

## 1 Introduction

Magnetic Holes (MHs) were first reported by Turner et al. (1977), and were defined as localized depressions in the magnetic field magnitude. Turner et al. (1977) also divided MHs into two categories, based on the rotation of the magnetic field across the MH: the linear magnetic holes, which have little to no rotation, and the rotational magnetic holes, which have a significant rotation. This division is commonly used, although the angular limit of what is classified as linear versus rotational varies. Studies that only consider linear MHs use limits varying from $10°$ to $25°$ (Madanian et al. (2020); Sperveslage et al. (2000); Tsurutani et al. (2011); Briand et al. (2010); Volwerk et al. (2020); Winterhalter et al. (1994); Xiao et al. (2010); Zhang et al. (2008)). Also, the temporal scales of MHs, regardless of their classification, have been reported to vary from a few seconds to several minutes (Karlsson et al. (2021); Madanian et al. (2020); Sperveslage et al. (2000); Turner et al. (1977); Volwerk et al. (2020); Winterhalter et al. (1994); Xiao et al. (2010); Zhang et al. (2008)).

Suggested formation mechanisms of linear MH range from being remnants of magnetic mirror mode structures (Winterhalter et al., 1994) to non-linear Alfvén waves (Tsurutani et al., 2002). The latter mechanisms include right-handed polarized Alfvén waves (Buti et al., 2001), and phase steepened Alfvén waves (Tsurutani et al., 2002). For the rotational MHs, the formation mechanism has been suggested to be related to slow magnetic reconnection (Turner et al., 1977), or remnants of solar coronal structures (Zurbuchen et al., 2001).

The mirror mode instability results from a strong ion temperature anisotropy and results in the generation of quasi-periodic decreases or peaks in the magnetic field strength. These structures have no velocity in the plasma frame but are convected with





the plasma velocity (Southwood and Kivelson (1993); Horbury et al. (2004)). If solar wind MHs are remnants of mirror mode structures, they are therefore expected to be convected with the solar wind. Horbury et al. (2004) also points out that if linear magnetic holes are related to mirror mode structures, we expect them to have a solenoid/cylindrical structure.

If linear MHs originate from Alfvén waves, they should move with a velocity of the order of the Alfvén velocity in the solar wind frame. The velocity distribution should also exhibit a double hump, representing structures traveling at plus or minus the
Alfvén velocity (Avinash and Zank, 2007).

Rotational MHs have been less investigated over the years. (Turner et al., 1977) connect the observations of rotational MHs with directional discontinuities (D-sheets) associated with a magnetic field decrease (Gosling, 2012), first observed by Burlaga (1968). The D-sheet is suggested to be the result of reconnection triggered by the large magnetic shear associated with the current sheath (Gosling, 2012). For larger rotational MHs (not kinetic scale), (Zurbuchen et al., 2001) suggest that they are
created from a magnetic flux tube in the corona reconnecting with an open magnetic field line. In both cases, the theory involves current sheet-like discontinuities.

An important open question regarding magnetic holes is determination of their velocities in the solar wind frame. This is important for two reasons:

1) Knowing the velocity, we can calculate the spatial scales of the MHs; furthermore, any uncertainty in the velocity deter-
mination will directly translate into spatial scale uncertainties.

2) As described above, some theories predict a non-zero velocity with respect to the solar wind. Determining the propagation velocities will be a test of these and other non-mirror mode theories.

Here we present the first statistical study investigating the velocity of solar wind MHs. This is done by applying multi-spacecraft timing techniques on Cluster data (Harvey, 1998). A similar method was used by Horbury et al. (2004) to investigate
the propagation velocity of the magnetic mirror mode structures in the magnetosheath. Below we will describe the methodology for such a velocity determination, show its results, and relate the derived velocities to the local Alfvén velocity and the magnetic field rotation of the magnetic holes.

## 2   Method

The Cluster mission consists of four spacecraft (S/C); C1, C2, C3, and C4. We have used observation from the first quarter
of 2005. In this period, the S/C separations were between 562 and 2552 km, and most of the MHs passed all 4 S/C with a large enough time difference that the timing method can be used. In this period, the S/C also had orbits with apogee in the solar wind, spending about 30% of the orbital time there. We use data from the following instruments: Fluxgate Magnetometer (FGM) (Balogh et al., 1997) for the magnetic field, used to identify the MH and the time difference between S/C. The full resolution of the magnetic field is 0.04 s. For ion velocity and density measurements, we used The Cluster Ion Spectrometry
(CIS) (Rème et al., 2001) with a resolution of 4 s, primarily the Hot Ion Analyser (HIA), but in the absence of HIA data, we used the Composition and Distribution Function analyser (CODIF). To calculate the Alfvén velocity we obtained the





electron density from the Waves of High frequency and Sounder for Probing of Electron density by Relaxation experiment (WHISPER)(Décréau et al., 1997) with a resolution of 2.2 s.

## 2.1 Event Identification

In this study, we are only interested in MHs located in the pristine solar wind. The solar wind regions were identified by first searching for regions where the cone angle ($arctan(|\sqrt{v_y^2 + v_z^2}/v_x|)$) of the velocity is smaller than 20°. To remove more turbulent regions, such as foreshock regions, we only consider MHs in regions where the normalized standard deviation of the magnetic field 30 s before and after the MH is less than 0.15. We define the normalized standard deviation as $B_{sd}/B_0$, where $B_{sd}$ is the moving standard deviation, calculated with a 300 s window, and $B_0$ is defined below. Since MHs will cause

an increase in the normalized standard deviation, the threshold is set low enough to filter out most of the turbulent regions but high enough for MHs to pass. As a result, some turbulent/foreshock regions are not excluded automatically. Such regions are often associated with high energy ion populations. By visually inspecting the energy spectrogram we removed these foreshock events. This visual inspection also ensures that we have correctly excluded magnetosheath intervals and only consider isolated MH and not wave trains, like mirror modes.

Figure 1 shows a MH centered in a 1 min window from 2005-03-26 observed by C1. Panel (a) shows the ion energy flux from the CIS-HIA instrument plotted on a logarithmic scale. Here we see the narrow distribution, typical for the solar wind. Panels (b) and (c) show the magnetic field magnitude and its components in GSE coordinates. The magnetic field decreases from ∼4 nT to ∼2 nT, during a short interval, otherwise staying relatively constant. The Y and Z components decrease from the background magnitude and return to a similar level after passing the MH, while the X component remains unchanged. This

example is a typical observation of a linear MH in the solar wind. In panel (d), the electron and ion densities are shown. The difference between the two arises from them being determined from two different methods and instruments; The ion density from HIA and electron density from WHISPER, the latter being more reliable. Panel (e) shows the ion temperature. The density and temperature increase inside the MH, suggesting total pressure balance, as has been reported before (e.g., Volwerk et al. (2020)). Panel (f) shows the ion velocity in GSE, and the S/C position is shown in panel (g).

In this paper, we have defined MHs as a structure fulfilling Equation 1, i.e., having at least a 50% reduction in magnetic field strength. This is similar to many other studies (e.g. Karlsson et al. (2021), Volwerk et al. (2020)). The background field $B_0$ is obtained using a 5 min sliding window. The relative change in **B** is smoothed using a 1 s long sliding window to remove high-frequency waves and noise.

$$B_{rel} = \frac{\Delta B}{B_0} = \left\langle \frac{|\mathbf{B}| - B_0}{B_0} \right\rangle_{1s} < -0.5 \tag{1}$$

$$B_0 = \langle |\mathbf{B}| \rangle_{300s} \tag{2}$$

We perform this process for all S/C. The last condition we impose is that the MH should be observed by all four S/C, which is determined by a visual inspection of the events.





## 2.2 Timing Analysis

We can determine the velocity of the MHs when localized events are found. We have applied the method described in (Harvey,
1998) and used it for Cluster observations of mirror modes in (Horbury et al., 2004). Sundberg et al. (2015) also applied a
similar technique for subproton scale MHs in the terrestrial plasma sheet.

As mentioned in the Introduction, it is believed that rotational MHs are sheet-like discontinuities, while linear MHs are
solenoid-like structures. The method we will use is typically used for a planar discontinuity. However, suppose magnetic holes
have a cylindrical shape. The timing method will then produce a normal in the plane containing the solar wind flow direction
and the magnetic field direction, as described by Horbury et al. (2004). Figure 2 (a) and (b) show an illustration of the geometry
of the S/C in relation to a linear MH. The plane perpendicular to the timing normal will then represent the minimum magnetic
field strength along a line perpendicular to the plasma flow. This also shows that events where the S/C passes the hole in
different depths can be used, as it will yield the same normal (Horbury et al., 2004).

Knowing the times $t_i$ and positions $r_i$ for the MH minimum for spacecraft $i$, we get a system of equations for the timing
normal vector $\hat{\mathbf{n}}_t$ (Equation 3), which is solved by introducing the 'slowness' vector $\mathbf{m}$ (Equation (4)), resulting in Equation 5.

$$v_t(t_j - t_i) = (\mathbf{r}_j - \mathbf{r}_i) \cdot \hat{\mathbf{n}}_t \tag{3}$$

$$\mathbf{m} = \frac{\hat{\mathbf{n}}_t}{v_t} \tag{4}$$

$$(t_j - t_i) = (\mathbf{r}_j - \mathbf{r}_i) \cdot \mathbf{m} \tag{5}$$

In principle, Equation 5 can be easily solved by picking one of the spacecraft as the reference spacecraft. A more symmetric
treatment, however, is to consider Equation 5 as an overdetermined system of equations and determine the structure velocity by
minimizing a quadratic error expressed in Equation 6. Here $\mathbf{N}$ is the total number of S/C, and $\mathbf{R}$ is the volumetric tensor, defined
in Equation 7. $t_{\alpha\beta} = t_\alpha - t_\beta$ is the time difference between S/C, determined by cross-correlation, and $k,l = x,y,z$, summation
over repeated indices is assumed. A detailed description of this method can be found in (Harvey, 1998). This method is not
valid when the S/C configuration is coplanar.

$$m_l = \frac{1}{2N^2} \left[ \sum_{\alpha=1} \sum_{\beta=1} t_{\alpha\beta}(r_{\alpha k} - r_{\beta k}) \right] R_{kl}^{-1} \tag{6}$$

$$\mathbf{R_{jk}} = \frac{1}{N} \sum_{\alpha=1}^{N} r_{\alpha j} r_{\alpha k} \tag{7}$$

$$\tag{8}$$

To determine the structure velocity in the plasma frame, we apply equation 9. Here $\mathbf{v_{sw}}$ is the solar wind velocity, determined
from HIA. From the timing method, we only obtain the velocity along the timing normal; thus, to find the velocity in the plasma
frame, we take the projection of the solar wind velocity along the timing normal.



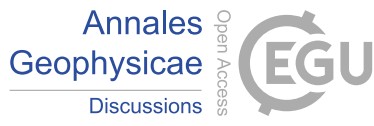

$$v_{pf} = v_t - \mathbf{v_{sw}} \cdot \hat{\mathbf{n}}_t \tag{9}$$

As addressed in many previous papers investigating the velocities of structures, there can be large uncertainties. We have applied a similar technique as Horbury et al. (2004) and Wang et al. (2020) to estimate the error. The two main sources of error are the determination of the time difference, and the solar wind velocity measurement (Knetter, 2005). For the time difference, we have taken the uncertainty to be two data points ($\sim 0.09$ s). There are six unique combinations of S/C with corresponding time differences. For each of these, we add two datapoints before and after, resulting in an array of five points for each S/C. This results in a total of 15625 unique combinations of S/C and time steps. We then find $v_t$ by taking the mean of all results, and the error is determined by the standard deviation. The velocity error in the solar wind frame is obtained by applying the error propagation formula. We have assumed that the error in the Y and Z components of the solar wind is negligibly small and that the error of the X component is 10%, similar to what was used in (Wang et al., 2020).

In this study we will investigate linear and rotational holes separately. Figure 2 c) shows the probability density function of the change of the magnetic field over the MH, $\Delta\phi$, for the events used in this study. The rotation is determined from the angle between the mean magnetic field 10 s before and 10 s after the MHs. The probability density is estimated using the kernel density estimation (KDE), a non-parametric way to derive the probability density of a univariate distribution (Rosenblatt, 1956). The distribution shows a clear peak at 13°, with a tail extending to angles approaching 180°. The vertical line at 50° indicates the boundary set for linear and rotational MHs, resulting in 40 linear and 29 rotational MHs. The boundary was set based on the distribution and by visually inspecting the MH with $25 \leq \theta \leq 60$. This angular distribution is similar to what has been shown in previous studies e.g., (Karlsson et al., 2021). The mean temporal scale size, determined by the full width at half minimum, of the 69 events was 5.9 s, with 5.4 s for linear and 6.8 s for rotational MHs

## 3 Results

Using the method described in section 2.1, we identified 72 MH events suitable for this study. However, only 69 will be used, as three were excluded due to the S/C configuration being close to coplanar and the timing method is not applicable. In appendix A, a table of all the events (including the three excluded ones) can be found, accompanied by the derived velocities and general parameters. Figure 3 panel (a)-(e) shows an example event of a linear MH from 2005-03-31, with a magnetic field rotation of 5°. Panel (a) shows the magnetic field magnitude in all four S/C, in Cluster colors (black, red, green, blue for C1-C4 respectively). We see that the MH is first seen by C4, followed by C3, C1, and C2. Panel (b) shows the same as (a) but shifted in time (cross-correlated) so that the magnetic minima are aligned. (c) shows the magnetic field components in GSE for S/C 4. Both the Y and Z components show a strong decrease, while the X component is more or less constant. (d) and (e) show the S/C positions in X-Y and X-Z projections, respectively, in units of $R_E$ (6371 km). For this event, there was a minimum and maximum S/C separation of 919 and 1333 km, respectively. In addition, the magnetic field direction is indicated by the black arrow, and the timing normal by the red one. The solar wind velocity measured by HIA for this event was $\mathbf{v}_{sw}$ = [-465 ± 47,





20, 9.8] km/s. The calculation of the timing normal using the method described above resulted in $\hat{\mathbf{n}}_t$ = [-0.93 ± 0.007 , -0.28 ± 0.018 , 0.24 ± 0.016]. The angle between the normal vector and the solar wind velocity was 22.7±0.4°, and the velocity in the normal direction $v_t = 432 \pm 8.76$ km/s. Using equation 9 we find the velocity in the solar wind frame to be $V_{pf} = 3.5 \pm 44$km/s. The errors are estimated as described in Section 2.2, and we can see that the error increases significantly for the velocity in the plasma frame. This is due to the high error of the solar wind velocity. For this event, the Alfvén velocity was 12.3 km/s.

We also show an example of a rotational MH from 2005-03-09 in Figure 3 panels (f)-(j), with the same format as panels (a)-(e). Here we can see a clear rotation of 137°of the magnetic field vector over the MH. For this event, there was a minimum and maximum S/C separation of 899 and 1076 km, respectively. The normal vector was found to be $\hat{\mathbf{n}}_t$ =[ -0.64 ± 0.017, -0.17 ± 0.014, 0.75 ± 0.015]. This is equivalent to an angle of 49 ± 1.6°with the solar wind velocity, $\mathbf{v}_{sw}$ = [ -680 ± 68, 46, 24] km/s. The velocity in the normal direction was 422 ± 7.9 km/s, and using equation 9 we determine the velocity in the plasma frame to be $v_{pf}$ = -12 ± 43 km/s. For this event, the Alfvén velocity was 60.1 km/s.

The results of the timing analysis applied to the whole dataset of MHs are shown in Figure 4. Panel (a) and (b) show the results for the linear, and (c) and (d) for the rotational MHs. The top panels show the MH velocities in the solar wind frame in the form of KDEs, and the bottom panel shows the MH velocity in the solar wind frame normalized with the Alfvén velocity. Considering the top panels first, we can see that the distributions for linear and rotational MHs are both centered close to zero. The median MH velocity in the solar wind frame is 7.4 ± 45 km/s and 25 ± 42 km/s for linear and rotational MH, respectively, while the mean is 19 km/s and 27 km/s. In Figure 4, the median and the mean Alfvén velocity are indicated with vertical lines, and the horizontal bar shows the error. For comparison, we obtain the mean velocity in the timing frame $v_t$ to be 436 ± 10 km/s.

To better see the relation between the MH velocity and the Alfvén velocity, the bottom panels show the MH velocity in the solar wind frame, normalized to the Alfvén velocity. The blue lines show $V_{pf}/V_A = \pm 1$, and the median and error are shown in red. The majority of the distribution lies within the Alfvén boundaries, with ∼ 78% and ∼ 67% for linear and rotational MHs respectively.

## 4 Discussion and conclusions

The level of uncertainty in this study is mainly determined by the uncertainties of the solar wind velocity. We have used an error of 10%, same as in Wang et al. (2020), in the x-direction and neglected the error in the other two components. As a result, the error will be larger for the events with higher solar wind velocity.

For the determination of the time shift, we choose a cross-correlation method. Another way would be to identify the minimum for all 4 S/C and directly take the difference. The linear MH shown in Figure 3 is a good example of why this method would not be a good fit for some events. In C4 one sees that it does not have the exact same shape as the other 3 S/C. The minimum is a bit displaced, which would result in the wrong time shift, an error which would propagate to the velocity and normal determination. This could also affect the determination from the cross-correlation, but we account for this in our error estimate.



Comparing the velocity to the local Alfvén velocity, we can see that the majority of the MHs, $\sim 70\%$ for both rotational and linear MHs, have velocities below the Alfvén velocity. We also see no sign of a double peak distribution, which we would expect if MHs move with the Alfvén velocity. The double peak distribution arises from structures moving with positive (towards Earth) or the negative (away from Earth) Alfvén velocity.

Our results are consistent with both rotational and magnetic holes convecting with the solar wind plasma. This is consistent with linear MH originating from mirror mode instabilities. Our results are comparable to what was found by Horbury et al. (2004) for mirror modes structures in the magnetosheath, where they also concluded that these structures are convected with the plasma. Wang et al. (2020) studied foreshock structures with a similar technique we used here. They found a clear peak centered around 100 km/s, indicating that the method used here can detect structures not convecting with the solar wind plasma. This supports our conclusion that MHs move with the solar wind.

For rotational MHs, theories suggest that they are the results of magnetic reconnection events and are similar to D-sheets, which should propagate with the solar wind. The rotational MHs have an median velocity of a factor of 2 larger than the linear ones, but still small enough to be consistent with convection with the solar wind.

Concluding that MHs are convected with the plasma, the next step will be to derive scale sizes since we can now confidently convert temporal scale sizes to spatial ones. In addition, the results presented here provide an estimate of the errors of the spatial scales as well. With the Cluster mission, we have identified several observations of MHs where the S/C passes different parts of the structure, which can be used to derive the limits of these scales. In the future, we plan to investigate the morphology of the MHs, starting with the linear MHs, by combining observations with different models, such as the one used by (Goodrich et al., 2021). As a follow-up study, the velocity of isolated MHs in the magnetosheath should be performed and compared with this study and the results of Horbury et al. (2004).

*Data availability.* Cluster measurements can be found through https://csa.esac.esa.int/csa-web/

*Author contributions.* HT initiated the study, performed the data analysis, and wrote the paper. All co-authors contributed to the analysis of the results and to the reviewing and editing of the paper

*Competing interests.* The authors declare that they have no conflict of interest.

*Acknowledgements.* We thank the Cluster team for providing data and support. We acknowledge the use of the ESA Cluster Science Archive. We are thankful for the ESA Archival Research Visitor Programme. We acknowledge the use of the irfu-matlab package found through https://github.com/irfu/irfu-matlab. We are also thankful for the useful discussions with the International Space Sciences Institute (ISSI)





team "Towards a Unifying Model for Magnetic Depressions in Space Plasmas". HT and TK are supported by the Swedish National Space
Agency (SNSA) grant 190/19. SR and TK are supported by the SNSA grant 90/17.

## Appendix A

Table A1: List of all MH used in this study with results, including identified outliers.

| yyyy | mm | dd | hh | mm | ss | $V_t$ | $V_{t,err}$ | $V_{sw}$ | $V_{pf}$ | $V_{pf,err}$ | $\Delta\phi$ |
|------|----|----|----|----|-----|-------|-------------|----------|----------|--------------|--------------|
| 2005 | 1 | 12 | 13 | 27 | 3,0 | 691,1 | 21,4 | 654,5 | 93,1 | 66,3 | 18,2 |
| 2005 | 1 | 15 | 1 | 45 | 30,7 | 439,3 | 9,2 | 606,8 | -39,5 | 52,0 | 40,1 |
| 2005 | 1 | 15 | 2 | 14 | 56,6 | 504,3 | 18,1 | 649,1 | 68,0 | 58,5 | 77,1 |
| 2005 | 1 | 15 | 2 | 21 | 36,6 | 301,8 | 4,2 | 628,0 | -73,9 | 48,1 | 148,2 |
| 2005 | 1 | 17 | 9 | 28 | 29,1 | 450,7 | 16,9 | 630,7 | -43,9 | 50,3 | 5,8 |
| 2005 | 1 | 28 | 21 | 7 | 9,3 | 398,4 | 6,8 | 387,2 | 25,4 | 35,8 | 8,1 |
| 2005 | 1 | 29 | 9 | 54 | 55,9 | 383,6 | 5,9 | 425,9 | -0,5 | 42,5 | 37,3 |
| 2005 | 1 | 30 | 14 | 30 | 53,9 | 431,3 | 8,3 | 580,3 | 13,5 | 45,9 | 14,4 |
| 2005 | 1 | 5 | 8 | 43 | 43,0 | 761,8 | 34,0 | 674,1 | 118,1 | 73,0 | 127,8 |
| 2005 | 2 | 10 | 0 | 10 | 22,7 | 400,9 | 6,2 | 682,1 | 13,5 | 44,1 | 5,6 |
| 2005 | 2 | 10 | 1 | 27 | 28,0 | 393,8 | 6,8 | 695,2 | 141,6 | 34,5 | 134,1 |
| 2005 | 2 | 10 | 3 | 28 | 18,3 | 605,8 | 24,9 | 698,0 | 52,3 | 53,8 | 9,6 |
| 2005 | 2 | 11 | 13 | 20 | 20,6 | 488,9 | 8,7 | 617,7 | -3,6 | 46,2 | 53,3 |
| 2005 | 2 | 14 | 13 | 30 | 21,1 | 286,9 | 3,6 | 389,8 | 63,3 | 24,4 | 118,2 |
| 2005 | 2 | 14 | 9 | 32 | 29,6 | 233,1 | 3,0 | 350,4 | -12,7 | 27,1 | 93,0 |
| 2005 | 2 | 17 | 3 | 41 | 9,0 | 393,4 | 4,9 | 384,8 | 36,3 | 37,5 | 1,5 |
| 2005 | 2 | 19 | 19 | 49 | 49,6 | 251,6 | 9,6 | 498,5 | 39,4 | 31,1 | 151,0 |
| 2005 | 2 | 19 | 19 | 57 | 41,2 | 458,1 | 21,2 | 509,1 | 0,1 | 45,5 | 3,0 |
| 2005 | 2 | 21 | 20 | 13 | 35,4 | 170,1 | 1,4 | 380,4 | -27,3 | 17,7 | 5,8 |
| 2005 | 2 | 21 | 21 | 10 | 35,6 | 370,2 | 4,5 | 375,9 | 5,0 | 37,4 | 0,7 |
| 2005 | 2 | 25 | 16 | 12 | 43,3 | 355,7 | 4,7 | 548,0 | 3,9 | 39,2 | 146,7 |
| 2005 | 2 | 25 | 18 | 31 | 59,9 | 135,1 | 0,6 | 523,0 | 36,1 | 8,9 | 52,9 |
| 2005 | 2 | 26 | 9 | 26 | 13,5 | 247,7 | 2,8 | 535,2 | -4,1 | 21,1 | 21,5 |
| 2005 | 2 | 28 | 21 | 56 | 55,4 | 633,7 | 15,6 | 614,9 | 56,6 | 60,5 | 78,3 |
| 2005 | 2 | 28 | 5 | 37 | 18,2 | 570,4 | 15,2 | 598,8 | 24,2 | 55,3 | 167,2 |
| 2005 | 2 | 2 | 18 | 40 | 17,4 | 354,3 | 4,8 | 501,4 | -8,0 | 39,9 | 58,7 |





| | | | | | | | | | | | |
|---|---|---|---|---|---|---|---|---|---|---|---|
| 2005 | 2 | 2 | 18 | 50 | 9,1 | 605,1 | 16,3 | 518,9 | 124,7 | 47,8 | 6,2 |
| 2005 | 2 | 2 | 19 | 37 | 26,9 | 515,2 | 10,6 | 519,1 | 3,6 | 52,0 | 37,5 |
| 2005 | 2 | 4 | 16 | 11 | 21,4 | 169,1 | 1,4 | 441,3 | 25,3 | 14,7 | 121,8 |
| 2005 | 2 | 7 | 18 | 59 | 30,4 | 530,3 | 9,4 | 670,7 | -6,0 | 60,7 | 4,8 |
| 2005 | 2 | 7 | 6 | 26 | 32,5 | 313,6 | 4,7 | 413,9 | 27,3 | 30,5 | 2,4 |
| 2005 | 2 | 9 | 18 | 24 | 58,0 | 648,6 | 18,9 | 685,3 | 28,0 | 63,8 | 0,8 |
| 2005 | 2 | 9 | 21 | 5 | 43,8 | 491,0 | 9,8 | 688,8 | 22,5 | 52,0 | 78,8 |
| 2005 | 2 | 9 | 23 | 46 | 44,2 | 667,6 | 18,1 | 729,5 | 5,9 | 66,9 | 103,1 |
| 2005 | 2 | 9 | 23 | 48 | 16,0 | 390,8 | 6,3 | 729,5 | 29,0 | 43,5 | 103,1 |
| 2005 | 2 | 9 | 2 | 38 | 57,9 | 521,7 | 10,7 | 725,6 | 35,3 | 51,2 | 117,8 |
| 2005 | 3 | 10 | 6 | 20 | 1,6 | 631,3 | 21,1 | 664,5 | 110,9 | 55,2 | 5,3 |
| 2005 | 3 | 10 | 6 | 56 | 41,5 | 337,8 | 5,2 | 670,4 | 44,8 | 29,7 | 109,7 |
| 2005 | 3 | 10 | 7 | 6 | 31,3 | 382,2 | 6,3 | 636,2 | 81,2 | 25,0 | 2,1 |
| 2005 | 3 | 12 | 1 | 19 | 27,4 | 244,7 | 2,2 | 435,3 | 0,7 | 22,5 | 24,8 |
| 2005 | 3 | 1 | 5 | 57 | 58,2 | 631,1 | 15,0 | 620,8 | 30,8 | 60,4 | 4,3 |
| 2005 | 3 | 23 | 19 | 31 | 11,2 | 335,0 | 3,9 | 355,1 | 24,2 | 30,1 | 12,5 |
| 2005 | 3 | 26 | 6 | 25 | 31,8 | 643,2 | 16,5 | 658,0 | 2,0 | 66,5 | 15,3 |
| 2005 | 3 | 2 | 18 | 43 | 30,0 | 635,6 | 22,4 | 607,3 | 62,3 | 59,6 | 21,4 |
| 2005 | 3 | 2 | 19 | 56 | 33,6 | 589,1 | 19,9 | 612,7 | 9,8 | 61,2 | 18,9 |
| 2005 | 3 | 31 | 5 | 36 | 19,0 | 432,0 | 8,7 | 464,8 | 3,5 | 44,0 | 5,0 |
| 2005 | 3 | 3 | 3 | 12 | 0,4 | 420,0 | 8,0 | 584,3 | -13,9 | 45,8 | 16,4 |
| 2005 | 3 | 5 | 23 | 43 | 58,8 | 229,1 | 2,6 | 392,5 | -6,1 | 22,7 | 1,4 |
| 2005 | 3 | 7 | 16 | 21 | 9,6 | 460,9 | 10,4 | 657,4 | -15,6 | 49,3 | 80,7 |
| 2005 | 3 | 7 | 16 | 23 | 6,1 | 662,3 | 26,8 | 657,4 | 48,3 | 68,9 | 81,0 |
| 2005 | 3 | 7 | 9 | 53 | 44,9 | 579,1 | 17,0 | 635,2 | -22,9 | 60,5 | 17,7 |
| 2005 | 3 | 9 | 17 | 2 | 59,2 | 351,8 | 5,1 | 699,2 | -24,5 | 41,5 | 149,9 |
| 2005 | 3 | 9 | 20 | 56 | 33,5 | 713,5 | 28,2 | 699,5 | 50,5 | 72,0 | 15,0 |
| 2005 | 3 | 9 | 21 | 24 | 10,6 | 421,9 | 8,0 | 672,7 | -12,3 | 43,4 | 136,5 |
| 2005 | 4 | 10 | 2 | 35 | 3,1 | 304,3 | 4,7 | 328,1 | -7,5 | 31,9 | 112,5 |
| 2005 | 4 | 10 | 5 | 0 | 14,7 | 184,3 | 1,5 | 322,1 | 8,5 | 18,9 | 83,1 |
| 2005 | 4 | 12 | 23 | 0 | 44,5 | 468,4 | 9,1 | 535,5 | -4,6 | 49,1 | 3,1 |
| 2005 | 4 | 14 | 13 | 26 | 19,1 | 412,5 | 8,0 | 517,0 | -85,2 | 50,6 | 83,2 |
| 2005 | 4 | 14 | 23 | 4 | 3,3 | 416,1 | 7,4 | 501,5 | -24,7 | 44,2 | 37,2 |

| 2005 | 4 | 15 | 1 | 45 | 2,6 | 258,6 | 3,6 | 477,9 | 23,2 | 25,1 | 13,9 |
|---|---|---|---|---|---|---|---|---|---|---|---|
| 2005 | 4 | 24 | 12 | 30 | 29,7 | 428,1 | 9,5 | 541,6 | 14,5 | 42,5 | 15,9 |
| 2005 | 4 | 24 | 9 | 22 | 56,8 | 330,5 | 6,2 | 532,8 | 50,7 | 25,8 | 8,3 |
| 2005 | 4 | 29 | 9 | 49 | 55,0 | 297,0 | 4,0 | 334,3 | -37,1 | 33,5 | 1,6 |
| 2005 | 4 | 29 | 9 | 54 | 3,9 | 437,6 | 8,6 | 334,1 | 106,0 | 34,1 | 5,8 |
| 2005 | 4 | 4 | 22 | 14 | 21,3 | 606,8 | 16,3 | 598,4 | 127,2 | 52,1 | 91,3 |
| 2005 | 4 | 5 | 11 | 22 | 18,7 | 357,4 | 5,4 | 639,1 | 33,8 | 34,9 | 137,6 |
| 2005 | 4 | 5 | 22 | 0 | 48,4 | 609,3 | 16,2 | 629,2 | -14,7 | 63,0 | 5,3 |
| 2005 | 4 | 7 | 23 | 1 | 45,2 | 375,0 | 5,4 | 402,9 | 4,0 | 36,4 | 3,2 |
| 2005 | 4 | 7 | 4 | 52 | 5,5 | 356,4 | 5,1 | 470,4 | 81,9 | 25,0 | 132,7 |
|  |  |  |  |  |  |  |  |  |  |  |  |
| Outliers |  |  |  |  |  |  |  |  |  |  |  |
| 2005 | 2 | 8 | 0 | 35 | 22,1 | 507,9 | 174,4 | 737,4 | 337,3 | 305,2 | 140,3 |
| 2005 | 2 | 8 | 0 | 37 | 0,0 | 559,3 | 160,7 | 737,4 | 270,2 | 299,7 | 141,4 |
| 2005 | 2 | 8 | 0 | 47 | 44,6 | 523,6 | 253,8 | 740,2 | 219,9 | 383,2 | 10,2 |

All velocities (V*) are in units of km/s and $\Delta\Phi$ is in the unit of degrees.



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



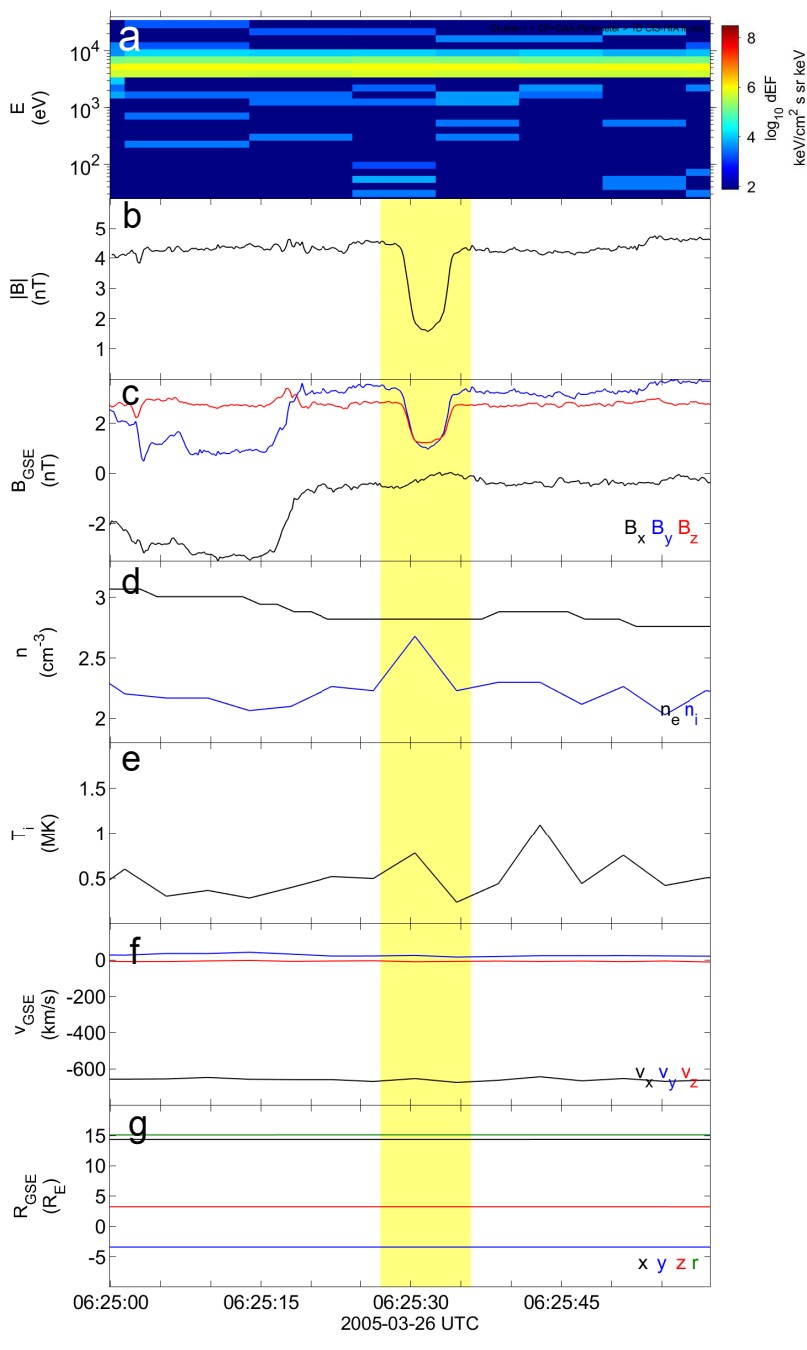

**Figure 1.** General parameters of a linear MH observed by C1. (a) Ion energy spectrum (HIA), (b) magnetic field magnitude (FGM), (c) C1 magnetic field components (FGM), (d) electron (WHISPER) and ion density (HIA), (e) ion temperature (HIA), (f) velocity (HIA) and (g) position in GSE (AUX). The yellow shaded area indicates the MH region.





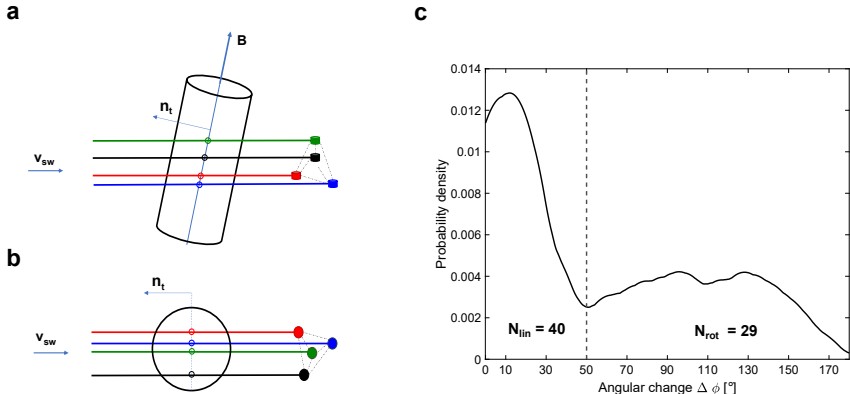

**Figure 2.** Panel (a) gives a 3D illustration of the configuration between MH and S/C. b) shows the 2D view from the top. The colors refer to typical Cluster S/C colors; black, red, green and blue for C1, C2, C3 and C4. Figure adapted from (Horbury et al., 2004). Panel c) probability density distribution of the change of the magnetic field across the MH. The vertical line indicates the limit for rotational and linear MHs.

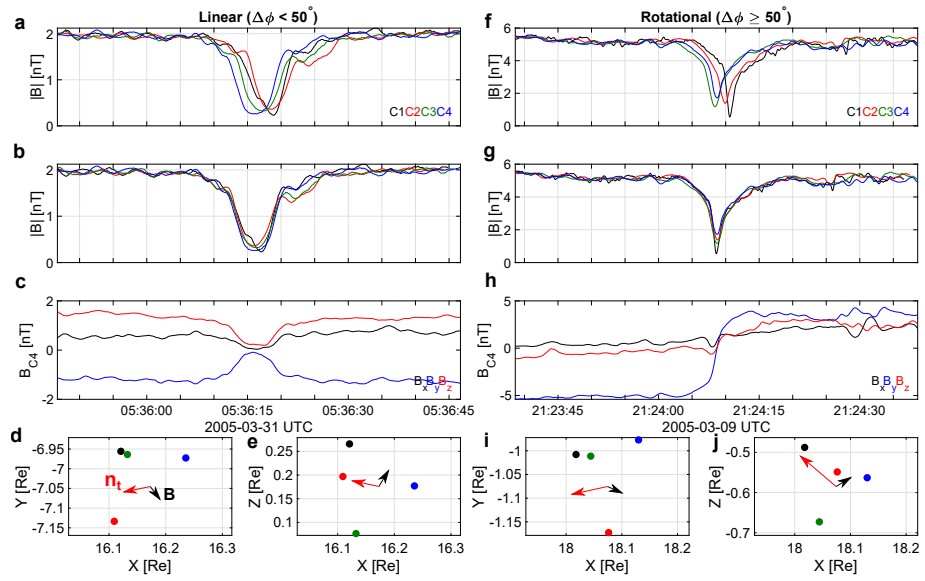

**Figure 3.** Example event of a linear (a)-(e) and a rotational (f)-(j) MH. Panel (a) shows the magnetic field magnitude observed in all S/C. (b) shows the same as (a) but with the S/C time shifted according to C4. (c) Shows magnetic field observed by C4. "S/C position in XY plane is given in panel (d), while (e) shows the position in XZ, both in GSE coordinate system. Both the last panels have the magnetic field and timing normal plotted in black and red respectively. Panels (f)-(j) have the same format as (a)-(e).





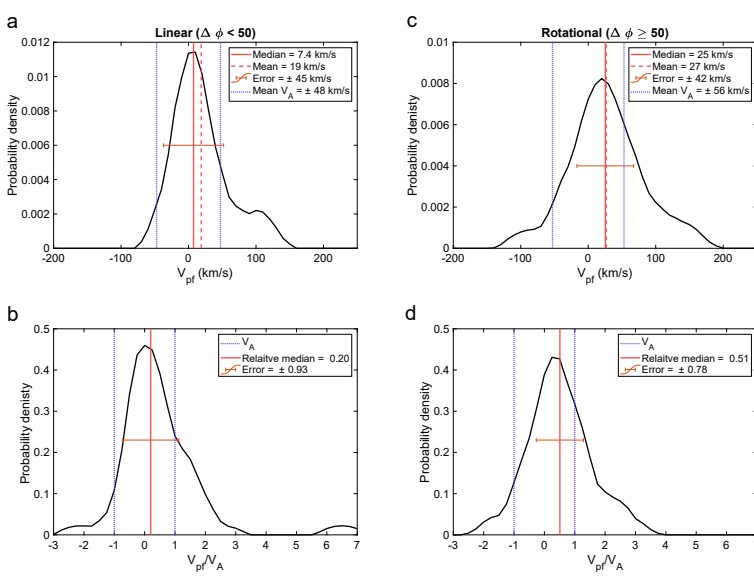

**Figure 4.** Probability density of the results, (a)-(b) for linear MHs and (c)-(d) for the rotational MHs. (a) and (c) show the results as a function of the MH velocity in the plasma frame. (b) and (d) show the results for the MH velocity relative to the mean Alfvén velocity.