# Peer review of "Velocity of magnetic holes in the solar wind from Cluster multipoint measurements"

_Annales Geophysicae, 2023_

## Referee Comment (RC1)

Referee report: Velocity of magnetic holes in the solar wind from cluster multipoint measurements\

Authors: Trollvik, Karlsson and Raptis

This simple and short paper deals with an important, and yet neglected (after 22 years of Cluster) topic, on whether magnetic holes are convected with the solar wind plasma or if they have an intrinsic velocity in the plasma frame. To do this the authors use the well-known timing-method by Harvey, which has proven itself well over the duration of the Cluster mission, on the magnetic field data. The solar wind velocity is taken from the CIS-CODIF/HIA instruments. The combination of the timing velocity and the solar wind velocity projected in the timing normal delivers then the hole-velocity in the plasma frame. Also, the Alfvén velocity is determined for each event, in order to check the possible creation of the holes. It is found that the magnetic holes are basically convected with the plasma flow and have only a small, sub-Alfvénic velocity in the plasma frame, where the error in the velocity is mainly determined by the error in the plasma instrument.

The paper is well-written, and the results are clear. However, there are a few minor points that should be checked:

- Page 2: Here the FGM is described and the text says that the full resolution of the instrument is 0.04 s, which would be roughly 25 Hz. However, the normal-mode of the Cluster FGM is 22 Hz. Then later on page 7 it is said that the error in the time is taken to be two data points, claimed to be ~0.09 s, but that does not agree with the 0.04 s on page 3. Therefore, it is better to write the sampling frequency, mentioning the ~0.09 s would then be okay.
- Page 3: "standard deviation" I think the authors have mixed up "low" and "high". To use the std to filter out the "noise" the threshold has to be high enough (put infinity, no signal comes through), but is should be low enough to let the real signal (the MH) through.
- Figure 1: I think panel g is unnecessary, the location can be put into the caption.
- Page 4, line 90: (Horbury et al., 2004) LaTeX \citep should be \citet
- Page 4, line 93: "However, suppose …", I think is would be better to write "However, we suppose that …"
- Page 4, line 96: "The plane perpendicular …" I do not understand this sentence. Timing analysis assumes that the structure is a plane wave moving over the tetrahedron, and one gets the normal to this plane. How this is related to "minimum magnetic field strength" is unclear to me. Do the authors mix up here maybe minimum variance of the magnetic field?
- Page 4, Eq. 6: There is no upper limit for the summation, and there is an empty Eq. 8
- Page 5: (Wang et al., 2020) LaTeX \citep should be \citet
- Page 6, line 129: The authors use an angle of 50° for the split between linear and rotating magnetic holes and base that on Karlsson et al. (2021). However, looking at that paper I seem to find that there the distribution of rotation angles shows a (inverted) knee at roughly 30° (their figure 4 and equations 6 and 7). What is the reason for choosing this angle differently? Clearly, the distribution shown in Fig. 2c indicates a boundary at 50, but how should that be combined with earlier determined boundaries?
- Page 5, line 143: "the magnetic field direction", I would add "background"
- Page 6, line 150: "to the high error" I would write "to the assumed high error"
- Page 7, line 189: "a factor of 2", this is more that a factor of 3 (25/7.4)

- Table A1: This table give a lot of information that can be used. I would ask for an additional 2 columns, one with the Q-value of the tetrahedron (see Robert et al., 1998, in "Analysis Methods for Multi-Spacecraft Data", Paschmann & Daly) and the angle of $\mathbf{n}_{timing}$ and $\mathbf{V}_{sw}$. The former will show an indication of how well the timing analysis is done and the latter will (maybe) show that current sheets (strong rotational holes) have a normal more tilted wrt. the solar wind direction.

---

## Author Comment (AC1)

**Response reviewer 1**

Firstly, we would like to thank the reviewer for useful comments and feedback. Below you will find each comment along with our responses in red text.

This simple and short paper deals with an important, and yet neglected (after 22 years of Cluster) topic, on whether magnetic holes are convected with the solar wind plasma or if they have an intrinsic velocity in the plasma frame. To do this the authors use the well-known timing-method by Harvey, which has proven itself well over the duration of the Cluster mission, on the magnetic field data. The solar wind velocity is taken from the CIS-CODIF/HIA instruments. The combination of the timing velocity and the solar wind velocity projected in the timing normal delivers then the hole-velocity in the plasma frame. Also, the Alfvén velocity is determined for each event, in order to check the possible creation of the holes. It is found that the magnetic holes are basically convected with the plasma flow and have only a small, sub-Alfvénic velocity in the plasma frame, where the error in the velocity is mainly determined by the error in the plasma instrument. The paper is well-written, and the results are clear. However, there are a few minor points that should be checked:

- Page 2: Here the FGM is described and the text says that the full resolution of the instrument is 0.04 s, which would be roughly 25 Hz. However, the normal-mode of the Cluster FGM is 22 Hz. Then later on page 7 it is said that the error in the time is taken to be two data points, claimed to be ~0.09 s, but that does not agree with the 0.04 s on page 3. Therefore, it is better to write the sampling frequency, mentioning the ~0.09 s would then be okay. Yes, this is a mistake from our part, and will be corrected in the manuscript.

- Page 3: "standard deviation" I think the authors have mixed up "low" and "high". To use the std to filter out the "noise" the threshold has to be high enough (put infinity, no signal comes through), but is should be low enough to let the real signal (the MH) through. – Yes, we have mixed up the two, and will change it in the paper.

- Figure 1: I think panel g is unnecessary, the location can be put into the caption. We agree with this, and will remove the last panel, and instead put location as text in the figure.

- Page 4, line 90: (Horbury et al., 2004) LaTeX \citep should be \citet – We will rephrase the text in the paper to "(..) and Horbury et al. (2004) used it for Cluster observations of mirror modes.

- Page 4, line 93: "However, suppose …", I think is would be better to write "However, we suppose that …". We suggest changing it to "Suppose, however, …"

- Page 4, line 96: "The plane perpendicular ..." I do not understand this sentence. Timing analysis assumes that the structure is a plane wave moving over the tetrahedron, and one gets the normal to this plane. How this is related to "minimum magnetic field strength" is unclear to me. Do the authors mix up here maybe minimum variance of the magnetic field? What is meant by this, is that for a cylindrical geometry, the plane that is relevant is a plane perpendicular to the flow velocity, and defined by the magnetic field minimum along each flow line. This methodology was also used by Horbury et al. (2004) when performing timing analysis on mirror mode structures.

- Page 4, Eq. 6: There is no upper limit for the summation, and there is an empty Eq. 8 The upper limit should be N, and the empty equation 8 has been removed.

- Page 5: (Wang et al., 2020) LaTeX \citep should be \citet. We will rephrase the text to "(..) was used by Wang et al. (2020)"

- Page 6, line 129: The authors use an angle of 50° for the split between linear and rotating magnetic holes and base that on Karlsson et al. (2021). However, looking at that paper I seem to find that there the distribution of rotation angles shows a (inverted) knee at roughly 30° (their figure 4 and equations 6 and 7). What is the reason for choosing this angle differently? Clearly, the distribution shown in Fig. 2c indicates a boundary at 50, but how should that be combined with earlier determined boundaries? We believe that for our data set the boundary at 50 degree corresponds to a distinctive change in the distribution. In previous studies, several different definitions have been used, so there is no standard definition to relate to. Also, the data are available in Appendix A, and can be regrouped by other boundaries, if desired.

- Page 5, line 143: "the magnetic field direction", I would add "background". We agree and will add it to the text.

- Page 6, line 150: "to the high error" I would write "to the assumed high error". Yes, we agree to this and will change it.

- Page 7, line 189: "a factor of 2", this is more that a factor of 3 (25/7.4) Yes, that is right, and we will change it in the text.

- Table A1: This table give a lot of information that can be used. I would ask for an additional 2 columns, one with the Q-value of the tetrahedron (see Robert et al., 1998, in "Analysis Methods for Multi-Spacecraft Data", Paschmann & Daly) and the angle of n_timing and V_sw. The former will show an indication of how well the timing analysis is done and the latter

will (maybe) show that current sheets (strong rotational holes) have a normal more tilted wrt. the solar wind direction. A good suggestion, we will add these values to the table.

Horbury, T. S., et al. "Motion and orientation of magnetic field dips and peaks in the terrestrial magnetosheath." *Journal of Geophysical Research: Space Physics* 109.A9 (2004).

---

## Author Comment (AC2)

**Response reviewer 2**

Firstly, we would like to thank the reviewer for useful comments and feedback. Below you can find the comments with our response in red.

This manuscript considers the dynamics and propagation velocity of magnetic holes observed in the solar wind. Magnetic holes are now common structures in space and heliospheric plasmas. They have been observed within the magnetosheath, magnetotail, and in the solar wind upstream of Earth and other planets. They have also been detected to accompany switchback events at small heliocentric distances. As such, these structures have high scientific importance for plasma heating and the dynamics of the solar wind plasma. This manuscript is well-written and has taken an interesting approach to perform a statistical study of these structures through using data from the Cluster mission collected during an early phase of the mission.
However, there are aspects in this work that need further consideration and improvement, and the paper has the potential to make important contributions after additional work and resubmission. These points are listed below:

The authors track the minimum magnetic field point to estimate the velocity of the structures. Tsurutani et al. (2011, cited in the manuscript) suggested that magnetic depressions (or magnetic holes) can change in size as they propagate. Particularly, rotational holes that are created as a result of reconnection in the solar wind can be very dynamic. Once the reconnection begins, it can continue to evolve for long periods and internal structures of the reconnection process can continuously change the boundary and the pressure balance. Yao et al. (2020, not cited) also showed that the magnetic dips can be expanding or contracting. Have the authors considered the possibility of contracting and/or expanding structures?
It is unlikely that the MHs will change considerably on these time scales, see Figure 3. In any way, the timing is mainly determined by the position of the minimum. The events presented in Yao et al. were observed in the magnetosheath, and are thus very susceptible to errors, due to the turbulent nature of the surrounding plasma. Also, the small separation of the spacecraft makes the timing very sensitive to small-scale variation within the structure in their study. (This is the reason why we did not include time intervals with very small spacecraft separations in our statistical analysis.)

Yao, S. T., Hamrin, M., Shi, Q. Q., Yao, Z. H., Degeling, A. W., Zong, Q.-G., et al. [2020]. Propagating and dynamic properties of magnetic dips in the dayside magnetosheath: MMS observations. Journal of Geophysical Research: Space Physics, 125, e2019JA026736. https://doi.org/10.1029/2019JA026736

Line 24: "... These structures have no velocity in the plasma frame but are convected with..."

How does this statement fits with the goal of this paper? It looks like to be the answer to the open question identified in line 37: "An important open question regarding magnetic holes is determination of their velocities in the solar wind frame." Perhaps you can add

in line 24 that "some studies" have shown that MH are stationary in the plasma rest frame.

Here we refer to mirror modes. It is still not clear if Magnetic Holes (MHs) are remnants of mirror modes, and thus that is part of the objective of this paper. If MHs are convected with the solar wind, as our work suggests, it supports the theory that they are related to mirror modes.

Line 70: I suggest moving the discussion for magnetic hole event selection before discussing Figure 1. As it currently reads, it seems that the MH events are identified visually. We agree with the reviewer's suggestion and we will move the discussion accordingly.

Line 77: The solar wind is a quasi-neutral flow. In the pristine solar wind, any real physical differences between electron and ion densities are immediately restored. As authors indicated the difference in measured densities are instrumental. However, the same plasma density should be used to calculate derived parameters, (i.e. Alfven speed, etc.). Can you comment on why you introduce these different instruments, and which one is the ultimate source of the plasma density in the study?  If the two measurements are complementary to give a better time resolution that should be stated in the text. We agree with this. In the updated manuscript, we will remove the ion measurements to avoid confusion since the electron density determined by the WHISPER instrument is generally considered to be quite reliable (e.g. Trotignon et al., 2001). WHISPER measurements were used for the calculation for the Alfvén velocity (line 56).

Also Line 77: "the latter being more reliable."
This should probably be double checked. In Fig. 1, the electron density (black line) seems to be flat within the MHs, while ion densities increase, a typical signature in MHs. The discrepancy shown in Fig 1 is most likely due to the different time resolution between the density measurements/instruments.

Line 82-87:  Was it also a requirement for all 4 s/c to show a similar magnetic depression levels? It is also possible that different s/c might cross different parts of the solenoid/cylindrical structure of the MH. Can you comment on that?  It is not a requirement, however, the separation of the S/C in 2005 was typically small compared to the MHs.  A few of the events have lower depression levels, suggesting that each S/C probes different parts of the MH. See figure 3 for S/C geometry related to structure. Investigating these types of crossing will be part of the continued work.

Line 96-97: "The plane perpendicular to …." This sentence does not make sense as written. A plane cannot represent the field strength. Please rephrase. Yes, we realize that this was not very clear. We will rewrite it, so that it is clearer why the method described can be applied to cylindrical structures as well as planar. See response to reviewer 1.

Line 10137:  Is t_alpha,beta influenced by the size of the averaging (sliding) window discussed in line 82? The averaging is meant to improve the cross-correlation, as small-scale variations will not affect the result.

Line 107: What were the conditions for cross-correlation? Please comment of you applied a certain threshold for the correlation. It would be interesting to see how this cross-correlation limits the number of events and/or the level of error in determining the velocity. A paragraph describing this would be a god addition to the paper and helps to justify the importance of your conclusions.
The events chosen were completely isolated, and thus the cross correlation was always very high, ref. fig 1 and 3. We will add a sentence about this in the Method section.

Line 135: The number of events seems very small to do a statistical analysis. Is it possible to extent the period of study? At a later phase of the mission, the s/c trajectories moved to cover the solar wind, and all 4 s/c were still able to measure the magnetic field. If lack of plasma measurements is an issue, one possibility is to use the shifted omni data, for instance, to determine the Alfven speed. We wanted the S/C separation large enough to obtain a well-defined timing result, but not large enough so that difference in minimum magnetic field strength is not too large.

Line 163: What is the significance of the mean velocity in the timing frame? It depends on the solar wind velocity and normal direction. Related to this, can the authors comment whether based on this study, these structures are still to be considered pressure balanced? If there is a perceived velocity in the plasma rest frame, meaning that the structure either is lagging or pushing forward, should this also cause a sort of asymmetry between the leading and trailing boundaries?
Our study gives a result on the velocity of the whole structure in the frame of the plasma. It is not clear that such a velocity (which we argue is consistent with zero) would affect the pressure balance. An asymmetry between the trailing and leading boundaries have not been considered in this study, but could of course conceivably affect the pressure balance. We believe such effects are more likely in the magnetosheath, where the pressure balance might be disturbed by the crossing of the bow shock. This will be the subject of a future study.

Trotignon, J. G., Décréau, P. M. E., Rauch, J. L., Randriamboarison, O., Krasnoselskikh, V., Canu, P., ... & Fergeau, P. (2001, September). How to determine the thermal electron density and the magnetic field strength from the CLUSTER/WHISPER observations around the Earth. In Annales Geophysicae (Vol. 19, No. 10/12, pp. 1711-1720). Copernicus GmbH.